Deep-learning convolutional neural networks with transfer learning accurately classify COVID-19 lung infection on portable chest radiographs

Kikkisetti Shreeja 1
Zhu Jocelyn 1
Shen Beiyi 2
Li Haifang 2
http://orcid.org/0000-0001-6403-2827 Duong Tim Q. 1 tim.duong@stonybrook.edu
1 Radiology, Montefiore Medical Center and Albert Einstein College of Medicine , Bronx, NY , USA
2 Radiology, State University of New York at Stony Brook , Stony Brook, NY , USA
Frauscher Ferdinand
Electronic publication date: 2020 Nov 5
Publication date: 2020
Volume: 8
Electronic Location ID: e10309
Received 2020 Sep 2; Accepted 2020 Oct 15
Copyright: © 2020 Kikkisetti et al.
Copyright year: 2020
Copyright holder: Kikkisetti et al.
License: This is an open access article distributed under the terms of the Creative Commons Attribution License, which permits unrestricted use, distribution, reproduction and adaptation in any medium and for any purpose provided that it is properly attributed. For attribution, the original author(s), title, publication source (PeerJ) and either DOI or URL of the article must be cited.
License URL: https://creativecommons.org/licenses/by/4.0/

Keywords: Coronavirus, Machine learning, Chest X-ray, Lung infection, Computed tomography

Funding: The authors received no funding for this work.

==============================
Portable chest X-ray (pCXR) has become an indispensable tool in the management of Coronavirus Disease 2019 (COVID-19) lung infection. This study employed deep-learning convolutional neural networks to classify COVID-19 lung infections on pCXR from normal and related lung infections to potentially enable more timely and accurate diagnosis. This retrospect study employed deep-learning convolutional neural network (CNN) with transfer learning to classify based on pCXRs COVID-19 pneumonia (N = 455) on pCXR from normal (N = 532), bacterial pneumonia (N = 492), and non-COVID viral pneumonia (N = 552). The data was randomly split into 75% training and 25% testing, randomly. A five-fold cross-validation was used for the testing set separately. Performance was evaluated using receiver-operating curve analysis. Comparison was made with CNN operated on the whole pCXR and segmented lungs. CNN accurately classified COVID-19 pCXR from those of normal, bacterial pneumonia, and non-COVID-19 viral pneumonia patients in a multiclass model. The overall sensitivity, specificity, accuracy, and AUC were 0.79, 0.93, and 0.79, 0.85 respectively (whole pCXR), and were 0.91, 0.93, 0.88, and 0.89 (CXR of segmented lung). The performance was generally better using segmented lungs. Heatmaps showed that CNN accurately localized areas of hazy appearance, ground glass opacity and/or consolidation on the pCXR. Deep-learning convolutional neural network with transfer learning accurately classifies COVID-19 on portable chest X-ray against normal, bacterial pneumonia or non-COVID viral pneumonia. This approach has the potential to help radiologists and frontline physicians by providing more timely and accurate diagnosis.

Introduction

Coronavirus Disease 2019 (COVID-19) is a highly infectious disease that causes severe respiratory illness (Hui et al., 2020; Lu, Stratton & Tang, 2020). It was first reported in Wuhan, China in December 2019 (Li et al., 2020c) and was declared a pandemic on Mar 11, 2020. The first confirmed case of coronavirus disease 2019 (COVID-19) in the United States was reported from Washington State on January 31, 2020 (Holshue et al., 2020). Soon after, Washington, California and New York reported outbreaks. COVID-19 has already infected 10 million, killed more than 0.5 million people, and the United States has become the worst-affected country, with more than 2.4 million diagnosed cases and at least 122,796 deaths (https://coronavirus.jhu.edu, accessed 28 June 2020). There are recent spikes of COVID-19 infection cases across many states and around the world and there will likely be second waves and recurrence.

A definitive test of COVID-19 infection is the reverse transcription polymerase chain reaction (RT-PCR) of a nasopharyngeal or oropharyngeal swab specimen (Tang et al., 2020; Wang et al., 2020). Although RT-PCR has high specificity, it has low sensitivity, high false negative rate, and long turn-around time (Tang et al., 2020; Wang et al., 2020) (currently ~4 days) although improvement and other tests are becoming available (CDC, https://www.cdc.gov/coronavirus/2019-ncov/lab/index.html). By contrast, portable chest X-ray (pCXR) is convenient to perform, has a fast turnaround, and is well suited for imaging contagious patients and longitudinal monitoring of critically ill patients in the intensive care units because the equipment can be readily disinfected, preventing cross-infection. pCXR of COVID-19 infection has certain unique characteristics, such as predominance of bilateral, peripheral, and low lobes involvement, with ground-glass opacities with or without airspace consolidations as the disease progresses. These characteristics generally differ from other lung pathologies, such as bacterial pneumonia or other viral (non-COVID-19) lung infection. Based on CXR and laboratory findings, clinicians might start patients on empirical treatment before the RT-PCR results become available or even if the RT-PCR come back negative due to high false negative rate of RT-PCR. Early treatment in COVID-19 patients is associated with better clinical outcomes. Similarly, computed tomography (CT), which offers relatively more detailed features (such as subtle ground-glass opacity (Li et al., 2020b; Xu et al., 2020)), has also been used in the context of COVID-19. However, CT suite and equipment are more challenging to disinfect, and thus it is much less suitable for examining patients suspected of or confirmed with contagious diseases in general and COVID-19 in particular. Longitudinal CT monitoring of critically ill patients in the intensive care units is also challenging. In short, pCXR has become an indispensable imaging tool in the management of COVID-19 infection, is often one of the first examinations a patient suspected of COVID-19 infection receives in the emergency room, and ideally used for longitudinal monitoring of critically ill patients in the intensive care units.

The usage of pCXR under the COVID-19 pandemic circumstances is unusual in many aspects. For instance, pCXR is preferred as it can be used at the bedside without moving the patients, but the imaging quality is not as good as conventional CXR. In addition, COVID-19 patients may not be able to take full inspirations during the examination, obscuring possible pathology, especially in the lower lung fields. Many sicker patients may be positioned on the side which compromises imaging quality. Thus, pCXR data under the COVID-19 pandemic circumstances are suboptimal and, thus, may be more challenging to interpret. Moreover, pCXR is increasingly read by non-chest radiologists in some hospitals due to increasing demands, resulting in reduced accuracy and efficiency.

Portable chest X-ray images contain important clinical features that could be easily missed by the naked eyes. Computer-aided methods can improve efficiency and accuracy of pCXR interpretations, which in turn provides more timely and relevant information to frontline physicians. Deep-learning artificial intelligence (AI) is increasingly used to analyze diagnostic images (Ehteshami Bejnordi et al., 2018; LeCun, Bengio & Hinton, 2015). AI has the potential to facilitate disease diagnosis, staging of disease severity and longitudinal monitoring of disease progression.

One common machine-learning algorithm is the convolutional neural network (CNN) (Krizhevsky, Sutskever & Hinton, 2012), which takes an input image, learns important features in the image such as size or intensity, and saves these parameters as weights and bias to differentiate types of images (Song et al., 2020). CNN architecture is ideally suited for analyzing images. Moreover, many of the machine learning algorithms are trained to solve specific tasks, where models need to be rebuilt from scratch if the feature changes. Transfer learning overcomes such drawback by utilizing knowledge acquired for one task to solve related ones. Transfer learning is useful when dealing with small sample size data because the pre-trained weights improve efficiency and performance (Byra et al., 2018; Samala et al., 2017).

Many AI algorithms based on deep-learning convolutional neural networks have been deployed for pCXR applications (Harris et al., 2019; Heo et al., 2019; Mekov, Miravitlles & Petkov, 2020) and these algorithms can be readily repurposed for COVID-19 pandemic circumstances. While there are already many papers describing prevalence and radiographic features on pCXR of COVID-19 lung infection (see reviews (Bao et al., 2020)), there are a few AI papers (Apostolopoulos & Mpesiana, 2020; Cohen et al., 2020; Elaziz et al., 2020; Hurt, Kligerman & Hsiao, 2020; Murphy et al., 2020; Ozturk et al., 2020; Pereira et al., 2020; Zhu et al., 2020a) to classify CXRs of COVID-19 patients from CXR of normals or related lung infections. The full potential of AI applications of pCXR under COVID-19 pandemic circumstances is not yet fully realized.

The goal of this pilot study is to employ deep-learning convolutional neural networks to classify normal, bacterial infection, and non-COVID-19 viral infection (such as influenza) against COVID-19 infection on pCXR. The performance was evaluated using receiver-operating curve (ROC) analysis. Heatmaps were also generated to visualize and assessment the performance of the AI algorithm.

Materials and Methods

Data sources

This retrospective study used publicly available pCXR of (i) COVID-19 infection, (ii) non-COVID-19 viral infection, (iii) bacterial pneumonia, and (iv) normal subjects. The COVID-19 pCXR were downloaded from on May 27th, 2020 (Cohen, Morrison & Dao, 2020). The original download contained 673 CT or pCXR images of COVID-19, SARS, acute respiratory distress syndromes, pneumocystis, streptococcus, legionella, Chlamydophila, E. coli, Klebsiella, lipoid, Varicella, and influenza. The labels for the data came from a metadata file associated with the open dataset. The final sample size for COVID-19 patients was 455 pCXR from 197 patients. We recognized that this dataset was a public, community-driven dataset and there are potential selection biases. A radiologist (B.S.) evaluated all images for quality and relevance and each case was COVID-19 positive based on available data. As a result of this evaluation, a few images that were deemed to be of poor quality, were excluded.

The other datasets were taken from the established Kaggle chest X-ray image (pneumonia) dataset (https://www.kaggle.com/paultimothymooney/chest-xray-pneumonia). Although the Kaggle database has a large sample size, we randomly selected a sample size comparable to that of COVID-19 to avoid asymmetric sample size bias that could skew sensitivity and specificity. The sample sizes chosen for bacterial pneumonia, non-COVID-19 viral pneumonia, and normal pCXR were 492, 552 and 532 patients, respectively. Similarly, a chest radiologist evaluated all images for quality.

CNN: A CNN, a type of neural network, is ideally suited for analyzing images. In a standard CNN model, a filter (window) travels over each region of an image and looks for different features such as edges, colorations, patches, and more in order to classify an image into a certain category. Our CNN architecture was based on VGG16 (Fig. 1), a convolutional neural network (Simonyan & Zisserman, 2014), architecture was utilized for computation efficiency and ease to implement, for immediate translation potential. Our VGG16 architecture had 13 convolution layers that each run a series of filters over the image to extract important features. The VGG16 model was used because it was pretrained on the ImageNet database and properly employs transfer learning which makes the training process efficient. In other words, instead of having to learn all the relationships in an image from scratch, the model is already familiar with that when transfer learning is employed. The data was normalized first by transforming all files into RGB images and resizing them into 224 × 224 pixels to make them compatible with the VGG16 framework. Next, the images were one-hot-encoded and randomly split into 75% training and 25% testing. One hot encoding means to turn all the categorical labels into numerical values containing zeroes and ones to make it much easier for the computer to read. VGG16 implements 13 convolutional layers: five Max Pooling layers and three Dense layers which sum up to 21 layers and 16 weight layers (Ren et al., 2020). Conv 1 has 64 filters while Conv 2 has 128 filters, Conv 3 has 256 filters while Conv 4 and Conv 5 have 512 filters. The first two layers have two sublayers while the 4th and 5th layers have three sublayers. A sublayer is another layer within a convolutional layer that further filters images and passes information down to the next sublayers. The information collected from all the sublayers is compiled and sent to the next layer to make a cohesive prediction. A max-pooling layer was used after each step in the model to down sample the input and identify its important features based on the methods described in Ren et al. (2020). A max-pooling layer reduces the dimensionality of the image and allows for assumptions to be made about features contained in the sub regions (Ren et al., 2020). All convolutional layers used rectified linear units (ReLUs) as an activation function because it adds a small number of learnable parameters (Ren et al., 2020). Three fully connected layers were used, each having 4,096 nodes. Fully connected layers compose some of the last few layers in a model and connect all the inputs from each layer to the activation unit of the next layer (Ren et al., 2020). Dropout layers were used, along with the Softmax function, to prevent overfitting. Dropout layers work by randomly setting the edges of hidden neurons to zero at each update of the training phase. The softmax function turns all the scores from the images into a normalized probability distribution, which helps make the final prediction (Ren et al., 2020). For data analysis, batch sizes of 32 were used to limit computational expense and trained for 50 epochs. Epochs can be thought of as iterations. Several optimizers were tested and Adams optimization function was found to yield the lowest validation loss. The learning rate was lowered from the recommended 0.01 to 0.001 to prevent overshooting the global minimum loss. Categorical cross entropy was used as a loss function since the loss value decreases as the predicted probability converges to the actual label.

Figure 1 VGG16 architecture.

VGG16 architecture with 16 weighted layers including three fully connected layers.

Convolutional neural network analysis was performed on the whole pCXR as well as virtually segmented lungs. Lung segmentation was performed using a CNN architecture with 22 convolutional layers, 4 max-pooling layers, and 4 merged layers for connectivity. A ReLu activation function was used with the Keras library. The output consisted of a mask of the segmented lungs. The segmented lungs were then fed into the CNN model for the Covid19 classification. This model was trained on the Montgomery dataset and achieved an IoU score of 0.956 and dice score of 0.972.

Heatmaps: To visualize the spatial location on the images that the CNN networks were paying attention to, heatmaps were generated with class activation maps algorithm (38). This was done by adding global average pooling into CNN and calculating gradient backpropagation given one specific output class to obtain the class activation maps, indicating the discriminative image regions CNN paid attention to.

Statistical methods and performance evaluation: Five-fold cross-validation was used for the test set separately. Performance of the prediction model used standard ROC analysis of the area under the curve (AUC), accuracy, sensitivity, specificity, precision, recall and F1 scores. Precision was computed using true positives divided by the sum of false positives and true positives; Recall was computed using the true positives divided by the sum of true positives and false negatives; F1 scores were the mean of recall and precision rates.

Results

Figure 2 shows examples of pCXR from a normal subject, patients with bacterial pneumonia, non-COVID-19 viral pneumonia, and COVID-19 infection. COVID-19 is often characterized by ground-glass opacities with or without nodular consolidation with predominance of bilateral, peripheral and lower lobes involvement. Non-COVID-19 viral pneumonia is often characterized by diffuse interstitial opacities, usually bilaterally. Bacterial pneumonia is often characterized by confluent areas of focal airspace consolidation.

Figure 2 CXR.

Examples of chest radiographs (A) normal, (B) COVID-19 viral pneumonia, (C) non-COVID-19 viral pneumonia, and (D) bacterial pneumonia. COVID-19 is often characterized by ground-glass opacities with or without nodular consolidation with predominance of bilateral, peripheral and lower lobes involvement. Non-COVID-19 viral pneumonia is often characterized by diffuse interstitial opacities, usually bilaterally. Bacterial pneumonia is often characterized by confluent areas of focal airspace consolidation. Arrows indicate regions of above-described characteristic features.

Figure 3 shows the training and validation loss and accuracy as a function of the epoch of the CNN models. Loss decreases and accuracy improved with increasing epoch for both training and validation dataset. The accuracy typically reached >0.8.

Figure 3 CNN training and validation.

CNN (A) training and (B) validation loss and accuracy. Loss decreases and accuracy improved with increasing epoch for both training and validation dataset.

CNN was used to classify COVID-19 pCXR from those of normal, bacterial pneumonia, and non COVID-19 viral pneumonia patients in a multi-class neural network model. The results of the multi-class CNN classification for the whole CXR in the form of the confusion matrix is shown in Table 1. The precision, recall, and F1 scores for the whole pCXR (Table 2) showed good to excellent performance. For CNN with transfer learning performed on the whole pCXR, the overall sensitivity, specificity, accuracy, and AUC were 0.79, 0.93, and 0.79, 0.84 respectively. For CNN performed on segmented lungs, the overall sensitivity, specificity, accuracy, and AUC were 0.91, 0.93, 0.88, 0.89 respectively. The performance was generally better using segmented lungs.

Table 1 Confusion table.

Confusion table showing the multiclass CNN classification (whole CXR).

	Normal	COVID-19	Non-COVID-19 viral pneumonia	Bacterial pneumonia	
Normal	122	3	17	2	
Covid19	6	102	3	6	
Non-COVID-19 viral pneumonia	16	2	94	20	
Bacterial pneumonia	4	1	30	85	

Table 2 Precision and recall rate and F1 score (whole CXR).

	Precision	Recall	F1-score	
Normal	0.82	0.85	0.84	
Covid19	0.94	0.87	0.91	
Non-covid19 viral pneumonia	0.65	0.71	0.68	
Bacterial pneumonia	0.75	0.71	0.73	

To visualize the spatial location on the images that the CNN networks were paying attention to for classification, heatmaps of the COVID-19 vs. normal pCXR are shown in Fig. 4. The CNN algorithm was able to localize the area of pathology on pCXR. For CNN performed on the whole pCXR, the majority of the hot spots were reasonably localized to regions of ground glass opacities and/or consolidations, but some hot spots were located outside the lungs. For CNN performed on segmented lungs, the majority of the hot spots were reasonably localized to regions of ground glass opacities and/or consolidations, mostly as expected. There were a few pixels outside the lung that the algorithm paid attention to. These “errors” could be due to small sample sizes. It learned from the training dataset and there may be information that the algorithm might consider important. Large sample size usually minimizes such “error.”

Figure 4 Heatmap.

pCXR from (A) a COVID-19 patient, (B) the corresponding segmented lung, (C) heatmap from CNN analysis using whole pCXR, and (d) heatmap from CNN analysis using segmented lung overlaid on whole CXR. Arrows indicated regions of ground glass opacity and/or consolidations.

Discussion

This study developed and applied a deep-learning CNN algorithm with transfer learning to classify COVID-19 CXR from normal, bacterial pneumonia, and non-COVID viral pneumonia CXR in a multiclass model. Heatmaps showed reasonable localization of abnormalities in the lungs. The overall sensitivity, specificity, accuracy, and AUC were 0.91, 0.93, 0.88, and 0.89 respectively (segmented lungs).

There are a few AI studies to date using machine learning methods to classify CXRs of COVID-19, normal and related lung infections. By the time this article is reviewed many more articles will be published. Hurt, Kligerman & Hsiao (2020) used a U-net CNN algorithm to predict pixel-wise probability maps for pneumonia on CXR on 10 COVID-19 patients. No ROC analysis was performed. Apostolopoulos & Mpesiana (2020) used deep-learning algorithm to predict COVID-19 CXR with 98.66% sensitivity, 96.46% specificity, and 96.78% accuracy from a collection of 1,427 CXRs of which 224 were COVID-19 CXRs. Elaziz et al. (2020) used an innovative feature selection algorithms and standard classifier to classify CXR between COVID-19 (N = 216) and non-COVID-19 (N = 1,675). This method achieved accuracy rates of 96.09% and 98.09% for each of the respective datasets. Note that patient cohorts were highly asymmetric. Murphy et al. (2020) used an AI to classify COVID-19 CXRs (N = 223) from non-COVID-19 CXRs (N = 231) with an 0.81 AUC and they also showed that AI outperformed expert readers. Ozturk et al. (2020) used an AI model to perform multiclass classification for COVID-19 (N = 127) vs. No-Findings (N = 500) vs. Pneumonia (N = 500) as well as a binary classification for COVID vs. No-Findings which achieved 87.02% and 98.08% accuracies, respectively. Pereira et al. (2020) performed a multiclass classification and a hierarchical classification for COVID-19 vs. pneumonia vs. no-finding using resampling algorithms, texture descriptors, and CNN. This model achieved a F1-Score of 0.65 for the multiclass approach and F1 score of 0.89 for the hierarchical classification. AUC and accuracy were not reported. AI has also been employed to stage pCXR disease severity against radiologist scores (Cohen et al., 2020; Zhu et al., 2020a). Our study had one of the larger cohorts, balanced sample sizes, and multi-class model. Our approach is also amongst the simplest AI models with comparable performance index, likely facilitate immediate clinical translation. Together, these studies indicate that AI has the potential to assist frontline physicians in distinguishing COVID-19 infection based on CXRs.

Heatmaps are informative tools to visualize regions that CNN algorithm pays attention to for detection. This is particular important given AI operates on high dimensional space. Such heatmaps enable reality checks and make AI interpretable with respect to clinical findings. Our algorithm showed that the majority of the hotspots were highly localized to abnormalities within the lungs, that is, ground glass opacity and/or consolidation, albeit imperfect. The majority of the above-mentioned machine learning studies to classify COVID-19 CXRs did not provide heatmaps. We also noted that CNN on whole pCXR image resulted in some hot spots located outside the lungs. CNN of segmented lungs solved this problem. Another advantage of using segmented lung is reduced computational cost during training. Transfer learning also reduced computational cost, making this algorithm practical. The performance is generally better using segmented lungs.

Most COVID-19 positive patients showed significant abnormalities on pCXR. Some early studies have even suggested that pCXR could be used as a primary tool for COVID-19 screening in epidemic areas (Ai et al., 2020), which could complement swab testing which still has long turnaround time and non-significant false positive rate. In some cases, imaging revealed chest abnormalities even before swab tests confirm infection (Fang et al., 2020; Li et al., 2020a). In addition, pCXR can detect superimposed bacteria pneumonia, which necessitates urgent antibiotic treatment. pCXR can also suggest acute respiratory distress syndrome, which is associated with severe negative outcomes and necessitates immediate treatment. Together with the potential widespread shortage of intensive care units and mechanical ventilators in many hospitals, pCXR may play a critical role in decision-making. A timely implementation of AI methods could help to realize the full potential of pCXR in this COVID-19 pandemic.

This pilot proof-of-principal study has several limitations. This is a retrospective study with a small sample size and the data sets used for training had limited alternative diagnoses. Although the Kaggle database has a large sample size for non-COVID-19 CXR, we chose the sample sizes to be comparable to that of COVID-19 to avoid asymmetric sample sizes that could skew sensitivity and specificity. Future studies will need to increase the COVID-19 sample size and include additional lung pathologies. The spatiotemporal characteristics on pCXR of COVID-19 infection and its relation to clinical outcomes are unknown. Future endeavors could include developing AI algorithms to stage severity, and predict progression, treatment response, recurrence, and survival, to inform and advise risk management and resource allocation associated with the COVID-19 pandemic, with inclusion of clinical variables in predictive models (Lam et al., 2020; Zhao et al., 2020; Zhu et al., 2020b).

In conclusion, deep learning convolutional neural networks with transfer learning accurately classify COVID-19 pCXR from pCXR of normal, bacterial pneumonia, and non-COVID viral pneumonia patients in a multi-class neural network model. This approach has the potential to help radiologists and frontline physicians by providing efficient and accurate diagnosis.

Additional Information and Declarations

Competing Interests

Author Contributions

Data Availability

The authors declare that they have no competing interests.

Shreeja Kikkisetti conceived and designed the experiments, performed the experiments, analyzed the data, prepared figures and/or tables, authored or reviewed drafts of the paper, and approved the final draft.

Jocelyn Zhu conceived and designed the experiments, analyzed the data, prepared figures and/or tables, authored or reviewed drafts of the paper, and approved the final draft.

Beiyi Shen conceived and designed the experiments, authored or reviewed drafts of the paper, and approved the final draft.

Haifang Li performed the experiments, analyzed the data, authored or reviewed drafts of the paper, and approved the final draft.

Tim Q. Duong conceived and designed the experiments, authored or reviewed drafts of the paper, and approved the final draft.

The following information was supplied regarding data availability:

The public dataset used is available at Kaggle: Joseph Paul Cohen, “COVID-19 Chest X-rays for Lung Severity Scoring.” Kaggle, 2020, DOI 10.34740/KAGGLE/DSV/1230802.

Image segmentation code is available at GitHub:

https://github.com/Shreejakikkisetti/covidclassification.

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
