# Peer review of "Deep-learning convolutional neural networks with transfer learning accurately classify COVID-19 lung infection on portable chest radiographs"

_PeerJ, doi:10.7717/peerj.10309_

## Round 0.1 · original submission · Minor Revisions

Please change the manuscript according to the reviewers' comments.

Thanks

Best regards,

Ferdinand

·

Basic reporting

no comment

Experimental design

It is unclear whether the labels they used originated from the metadata associated with the publicly available pCXR images. The authors mention that “a chest radiologist evaluated all [] images for quality”. Were any images filtered out or relabeled as a result of this step? Did the radiologist re-classify the images? If that’s the case, how does their input compare with the starting classification? Were there other factors that were used to diagnose each disease type? For example, in reference 19, were symptoms and/or RT-PCR used in conjunction with the examination of pCXR lung images?

The authors mention splitting their data set into training and testing. It would benefit the reader to know how, for example, did the authors use random sampling, stratified sampling, etc. ? In addition, was five-fold cross-validation implemented for the test set separately?

Validity of the findings

no comment

Additional comments

I have read the study entitled “Deep-learning convolutional neural networks with transfer learning 2 accurately classify COVID19 lung infection on portable chest 3 radiographs” by Kikkisetti et al. In this study, the authors develop a deep learning convolutional neural network able to classify pCXR lung images into 1) COVID-19 infections, 2) other viral infections, 3) bacterial infections, and 4) normal. They report encouraging quality metrics, specifically for the CNN that was trained on segmented lung pCXRs. I especially commend the authors for clearly outlining their methodology as well as the limitations of their study in the discussion section.

With that said, one minor comment I have regards the training data. It is unclear whether the labels they used originated from the metadata associated with the publicly available pCXR images. The authors mention that “a chest radiologist evaluated all [] images for quality”. Were any images filtered out or relabeled as a result of this step? Did the radiologist re-classify the images? If that’s the case, how does their input compare with the starting classification? Were there other factors that were used to diagnose each disease type? For example, in reference 19, were symptoms and/or RT-PCR used in conjunction with the examination of pCXR lung images?

The authors mention splitting their data set into training and testing. It would benefit the reader to know how, for example, did the authors use random sampling, stratified sampling, etc. ? In addition, was five-fold cross-validation implemented for the test set separately?

Reviewer 2 ·

Basic reporting

No comments

Experimental design

no comments

Validity of the findings

no comments

Additional comments

This manuscript by Kikkisetti et al. reported their study on Portable chest x-ray (pCXR) combined with deep-learning convolutional neural networks (CNN) to classify COVID-19 lung infections. Segmented lungs showed better performance compared to the whole pCXR. In general, the manuscript is clearly presented and written, and contains appropriate introductory material, methods, statistics and reasonable. As noted above, the entire study sounds entirely plausible and provides useful information for further COVID19 lung infection classify at this special period. However, there are several minor concerns as described as follows. I would suggest the paper being accepted for publication if the authors can incorporate these comments in their revision.
There are several concerns proposed as follows:

Minor concerns:
1. The abstract is very good and has clear logical analysis, but the results of the paper are less clearly described. For example, In the abstract, there are many data COVID-19 pneumonia (N=455), normal (N=532), bacterial pneumonia (N=492), and non-COVID viral pneumonia (N=552). But how these data used to analyze and got the results should clearly present in the RESULTS part and also labeled in the table legend. The results part not just describe the figures and tables, it should follow your research logics and provided the information why you did this experiment and how you did it and what result you got.
2. On line 205, “but some hot spots were located outside the lungs”, could you explain why this situation appears and how to improve this?
3. For Figure 2 and Figure 4, you just give some examples, that’s fine. But would you give a semiquantitative analysis to show how many samples are represented by the images in each figure?
4. You may move Figure 1 to supplemental materials.

Reviewer 3 ·

Basic reporting

Only minor english language use/punctuation issues e.g. lines 52,110,111,112.

In the few months since the manuscript was written, the number of US COVID-19 related deaths has almost doubled. In the revision this should be updated. Lines 55 and 56.

Lines 60-62 the authors state that PRC has a low sensitivity, high false negative rate and long turn-around time of 4 days. The authors should double check if this is uniformly still the case or is unique to the USA.

Experimental design

Description of deep learning convolutional neural networks. For readers not familiar with convolutional neural networks there needs to be more detail explaining how it all works. The authors also need to explain what the following jargon means:
VGG16 model
one-hot-encoded
convolutional layers and sublayers
filters
max-pooling
dropout layers
soft max function
etc, etc

Validity of the findings

No comment

Additional comments

This timely paper nicely demonstrates how with relative ease, CNN and be used on portable chest x-rays for assessment and analysis of a new disease when clinical experience is limited. This could change management in sick patients and be life saving.

---

## Round 0.2 · accepted · Accept

This was a well-performed revision.

Thanks

Best regards

Ferdinand